# Effect of Methionine Diet on Time-Related Metabolic and Histopathological Changes of Rat Hippocampus in the Model of Global Brain Ischemia

**DOI:** 10.3390/biom10081128

**Published:** 2020-07-30

**Authors:** Maria Kovalska, Petra Hnilicova, Dagmar Kalenska, Anna Tomascova, Marian Adamkov, Jan Lehotsky

**Affiliations:** 1Department of Histology and Embryology, Jessenius Faculty of Medicine, Comenius University in Bratislava, 03601 Martin, Slovakia; maria.kovalska@uniba.sk (M.K.); marian.adamkov@uniba.sk (M.A.); 2Department of Neuroscience, Biomedical Center Martin, Jessenius Faculty of Medicine, Comenius University in Bratislava, 03601 Martin, Slovakia; petra.hnilicova@uniba.sk (P.H.); anna.tomascova@uniba.sk (A.T.); 3Department of Anatomy, Jessenius Faculty of Medicine, Comenius University in Bratislava, 03601 Martin, Slovakia; dagmar.kalenska@uniba.sk; 4Department of Medical Biochemistry, Jessenius Faculty of Medicine, Comenius University in Bratislava, 03601 Martin, Slovakia

**Keywords:** rat, hippocampus, neurodegeneration, brain ischemia and reperfusion, methionine diet, ^1^H MRS

## Abstract

Hyperhomocysteinemia (hHcy) represents a strong risk factor for atherosclerosis-associated diseases, like stroke, dementia or Alzheimer’s disease. A methionine (Met)-rich diet leads to an elevated level of homocysteine in plasma and might cause pathological alterations across the brain. The hippocampus is being constantly studied for its selective vulnerability linked with neurodegeneration. This study explores metabolic and histo-morphological changes in the rat hippocampus after global ischemia in the hHcy conditions using a combination of proton magnetic resonance spectroscopy and magnetic resonance-volumetry as well as immunohistochemical analysis. After 4 weeks of a Met-enriched diet at a dose of 2 g/kg of animal weight/day, adult male Wistar rats underwent 4-vessel occlusion lasting for 15 min, followed by a reperfusion period varying from 3 to 7 days. Histo-morphological analyses showed that the subsequent ischemia-reperfusion insult (IRI) aggravates the extent of the sole hHcy-induced degeneration of the hippocampal neurons. Decreased volume in the grey matter, extensive changes in the metabolic ratio, deeper alterations in the number and morphology of neurons, astrocytes and their processes were demonstrated in the hippocampus 7 days post-ischemia in the hHcy animals. Our results suggest that the combination of the two risk factors (hHcy and IRI) endorses and exacerbates the rat hippocampal neurodegenerative processes.

## 1. Introduction

Methionine (Met) is an essential amino acid present in food, which is regularly consumed within the Western diet [1,2,3,4,5]. The impact of a high and moderate Met diet on different tissues has been discussed in many studies [1,2,3]. An elevated level of Met in plasma, hypermethioninemia (hMet), has been linked with memory deficits and morphological changes in the hippocampus of young rats [6]. Chronic hMet and its oxidative product, Met sulfoxide, induces cellular oxidative stress [7,8] in various organs and also contributes to the brain pathology [9,10,11]. Moreover, Met can undergo spontaneous self-assembly to form amyloidogenic aggregates [12], and an increased Met oxidation in the apolipoprotein A-I could nucleate amyloidogenesis, which eventually leads to the aggregation into amyloid fibrils [13].

Furthermore, protein intake rich in Met content or dysregulation of Met metabolism within the “Met-homocysteine” cycle can concomitantly lead to the elevation of homocysteine (Hcy) in the circulating plasma. Hcy is a sulphur-containing amino acid produced by metabolic conversion of Met to cysteine. Its elevated level in plasma, hyperhomocysteinemia (hHcy), is one of the known risk factors for cardio- and cerebrovascular disorders. Clinical relevance of elevated total Hcy level in plasma has been proven by many studies on different tissues [14,15]. Recently, it was shown that neuroinflammation followed by a significant increase of cognitive deficits and micro-haemorrhages were manifested on an established model of hHcy in mice by a restricted intake of B vitamins and enriched Met diet [16]. In fact, cerebral microbleeds, as a result of chronic and acute pathology, has been detected by many authors, see for review [17,18,19]. Remarkably, development of the neurodegenerative disorders, such as progressive atherosclerosis and acute ischemic stroke, cognitive impairment, dementia, or Alzheimer’s disease (AD), were also shown to be associated with hHcy [1,3,5], but the exact mechanism of its involvement in an affecting neuronal tissue is not yet elucidated.

In our previous work on the rat hHcy model induced by a Met-enriched diet [20], we have revealed considerable cerebral dysregulations: (i) on the level of cerebral metabolites, and (ii) histologically detected acceleration of the hippocampal neurodegeneration. Previous papers from our laboratory [21,22,23,24] documented that the combination of hHcy induced by direct Hcy injection with the subsequent ischemia-reperfusion insult (IRI), aggravates neurodegenerative processes and might eventually lead to the development of AD-like pathology [22,23]. Using a novel methodological approach, we demonstrate here that the rats with the Met-enriched diet (inducing hHcy) were subject to the IRI manifested exacerbated patterns of hippocampal neurodegeneration. Our approach combines the detection of the metabolic ratio by the in vivo magnetic resonance spectroscopy (MRS) and magnetic resonance imaging (MRI) with the histo-morphological analysis, to explore the proposed aggregated impact of two risk factors on the affected brain region. 

## 2. Materials and Methods 

### 2.1. Induction of Ischemia-Reperfusion Injury (IRI) 

Our experiment was accomplished according to Directive 2010/63/EU of the European Parliament and the Council on the protection of animals used for scientific purposes. Animals used in this experiment were handled in accordance with the directive for Animal Care and Health of the State Veterinary and Food Department of the Slovak Republic (approval number 727/12-221) for animal experiments. 

Adult male Wistar rats (Velaz, Prague, Czech Republic) at the age of 5–6 months and weight of 300–400 g (mean body weight of 332 g, total *n* = 38) were used in this experiment. Animals were kept under the standard conditions (air-conditioned rooms with a temperature of 22 ± 2°) and 12 h day/night cycle. Water, as well as food, were available ad libitum. Global IRI was evoked by the 4-vessel occlusion (4-VO) model in accordance with Pulsinelli et al. [25]. In brief, rats were anaesthetized with 4.5% sevoflurane in a mixture of 33% O_2_ and 66% N_2_O. To sustain the anaesthesia throughout the operation, 3–3.5% sevoflurane was used. Both vertebral arteries were irreversibly occluded through the alar foramina by thermocoagulation on day one. Furthermore, on day two, both common carotid arteries were occluded for 15 min by small atraumatic clips while preserving the same anaesthetic conditions as mentioned above. The rats then underwent a reperfusion period for 3 and 7 days. Due to different analyses used in this study, the animals were divided into two separated groups. The first group of animals underwent histological analysis. This group was divided into 5 subgroups see 2.3 (*n* = 5/group). The second group of 8 animals was four times analysed by MR. Before Met diet (at day 0), at the end of Met diet (at day 28), 3 days after carotid occlusion and 7 days post-ischemia. Animals were sacrificed after a particular reperfusion period in accordance with ethical principles. Brains were rapidly dissected from the skull and processed for subsequent procedures. Control groups were processed in the same manner as mentioned above with the exception of carotid occlusion. 

### 2.2. Induction of Mild Hyperhomocysteinemia (hHcy) by Met Enriched Diet

Rats underwent Met diet in the duration of 28 days before the experiment. Met (L-methionine, Sigma-Aldrich, Taufkirchen, Germany) was given in drinking water at a dose of 2 g/kg of animal weight per day in accordance with Xu et al. [1]. The daily volume of water intake was established at 46.53 ± 8.34 mL for rats in all experimental groups. After this treatment, a mild hHcy was evoked in the animals. Determination of plasma Hcy concentration followed. At day 29, from the naïve controls and controls treated by the Met diet, peripheral blood samples (1.5 mL) were collected from the retro-orbital venous plexus. The blood was immediately cooled on ice and centrifuged. The supernatant was collected and plasma was stored at −80 °C. The plasma Hcy levels were measured by enzymatic assay with a Hcy Liquid Stable Reagent Kit (Axis-Shield Diagnostics, Dundee, Scotland) in accordance with the instructions of the manufacturer and analysed with an automatic biochemical analyser (Siemens ADVIA 1650).

### 2.3. Experimental Groups of Animals

The first group of animals was divided as follows:(1)control (naïve) animals (C, *n* = 5).(2)naïve animals that underwent 15 min ischemia and 3 days of reperfusion (IR-3d, *n* = 5).(3)naïve animals that underwent 15 min ischemia and 7 days of reperfusion (IR-7d, *n* = 5).(4)animals after 28 days with Met diet without ischemic insult–Met control (Met-C, *n* = 5).(5)animals after 28 days with Met diet that underwent 15 min ischemia and 3 days of reperfusion (Met-IR-3d, *n* = 5).(6)animals after 28 days with Met diet that underwent 15 min ischemia and 7 days of reperfusion (Met-IR-7d, *n* = 5).

Because of different handling and methods of analysis, the second group of animals was separated into independent group. This group of animals (*n* = 8) underwent the same procedures as subgroups 4 followed by the same process as in subgroup 6. 

### 2.4. FluoroJade-C Staining

All of the animals from group one (*n* = 5/subgroup) were put in an anaesthetic box and put to sleep by spontaneous inhalation of 3.5% sevoflurane in a mixture of oxygen and nitrous oxide (33/66%). Animals were consecutively trans-cardially perfused with 0.1 mol/l phosphate-buffered saline (PBS, pH 7.4) ensued by 4% paraformaldehyde in 0.1 mol/l PBS (pH 7.4). After perfusion, all animals were decapitated. Brains were dissected from the skull and immersed overnight in the same fixative at 4 °C. Afterwards, the rat brains were placed in 30% sucrose for the next 24 h at 4 °C. Lastly, the rat brains were embedded in embedding medium (Killik, Bio Optica, Milano, Italy). Embedded brains were immediately frozen by a fast cooling boost in a cryobar Shannon Cryotome E (Thermo Fisher Scientific, Waltham, MA, USA) and sectioned at 30 μm thick sections. The sections were arranged on Superfrost Plus glass (Thermo Fisher Scientific, Waltham, MA, USA). FluoroJade-C (FJC) was used as a marker of neurodegeneration. The brain sections mounted on the Superfrost Plus glasses were heated at 50 °C for 30 min before staining. After heating, the slides were immersed in absolute alcohol (3 min) followed by 70% alcohol (1 min) and distilled water (1 min). Sequentially, slides were immersed in a solution of 0.06% potassium permanganate for 15 min and rinsed in distilled water for 2 min. After 2 h in the staining solution, 3 × 1-min washes in distilled water followed. The slides were dried at room temperature and cover slipped with Fluoromount™ Aqueous Mounting Medium (Sigma-Aldrich, Taufkirchen, Germany) according to the standard protocols. 

### 2.5. Fluorescent Immunohistochemistry 

The primary antibody used in this work was GFAP (glial fibrillary acidic protein; 1:200; AB5804, Millipore, Darmstadt, Germany), which presents a specific marker for astrocytes. A marker of mature neurons, NeuN (Neural Nuclei; 1:100; 24307S; Cell Signalling Technology, Danvers, MA, USA) rabbit antibody was applied as well. 

Brain sections obtained from 1–6 subgroups (prepared as described above) were permeabilized with 0.1% Triton X-100, pre-blocked with 10% BSA (bovine serum albumin) for 60 min. Consecutively, the brain sections were incubated with O/N at 4 °C with use of a solution of primary antibodies diluted in the 0.1% Triton X-100 with 10% BSA supplementation. Immunofluorescent detection was carried out using Alexa Fluor 488 goat-anti-mouse IgG (A11001, 1:100, Life Technologies, Carlsbad, CA, USA)-conjugated secondary antibody and Alexa Fluor 594 goat-anti-rabbit IgG (A11012, 1:100, Life Technologies)-conjugated secondary antibody. Sections were mounted with Fluoromount-G^®^ medium containing DAPI (4′, 6-diamidino-2-phenylindole; CA 0100-20, SouthernBiotech, Birmingham, AL, USA) according to the standard protocols. No immunoreactivity was determined in the absence of the primary antibody. The brain slides were examined by a confocal laser scanning microscope, Olympus FluoView FV10i (Olympus, Tokyo, Japan) in the hippocampal cornu ammonis 1 (CA1) area. Cells were counted by a double-blind method performed by two observers on three random microscopic fields. The objective of 10× with zoom up to 40× magnification provided with filters for FITC (fluorescein isothiocyanate—Fluoro-Jade C; excitation: 495 nm; emission: 519 nm), Alexa Fluor 488 (excitation: 499 nm; emission: 520 nm) and Alexa Fluor 594 (excitation: 590 nm; emission: 618 nm) was applied. The image capture was accomplished with Olympus Fluoview FV10-ASW software, version 02.01 (Olympus), Quick Photo Micro software, version 2.3 (Promicra, Prague, Czech Republic) and processed in Adobe Photoshop CS3 Extended, version 10.0 for Windows (Adobe Systems, San Jose, CA, USA).

### 2.6. In Vivo Magnetic Resonance Examination

For the in vivo magnetic resonance (MR) examination, a 7 T Bruker BioSpec small animal MR scanner (Bruker BioSpin MRI, Ettlingen, Germany) was used. The ^1^H radio-frequency resonator (Bruker BioSpin MRI, Ettlingen, Germany) for RF transmission was utilized. For signal reception in brain areas, the 4-elements ^1^H surface array coil (Bruker BioSpin MRI, Ettlingen, Germany) was applied.

The second group of experimental animals, control (*n* = 8), Met-C (*n* = 8), Met-IR-3d (*n* = 5), and Met-IR-7d (*n* = 5) was anesthetized with 4% sevoflurane in a mixture with medical O_2_ (for induction of anaesthesia) and then with 2–4% sevoflurane for anaesthesia sustentation. Animals were stabilized with a tooth holder and nose mask in a specialized water heated bed (Bruker BioSpin MRI). Body temperature and respiratory rate were during the scanning procedure also monitored. 

To secure similar head positioning, two-dimensional T_1_-weighted reference images were obtained within 12.8 s. For the exact volume of interest (VOI), localization of the T_2_-weighted MRI in coronal, sagittal, and transversal planes were acquired with the two-dimensional turbo spin echo (RARE; rapid acquisition with relaxation enhancement) sequence. The following parameters were obtained: TR (repetition time)/TE (echo time) = 2680/40 ms, 23 slices with 0.5 mm thickness and 0.3 mm gap, 2 averages, RARE factor = 10, FOV (field of view) = 35 × 35 mm^2^, image size = 256 × 256, resolution = 0.137 × 0.137 × 0.5 mm^3^, and total acquisition = 2 min 14 s. For volumetry purposes, the T_2_-weighted two-dimensional turbo RARE MRI of the whole brain in coronal plane was determined with 25 sections (0.5 mm thickness and 0 mm gap). 

For identification of magnetic field (B_0_) inhomogeneity across the brain, B_0_ phase map was gained prior to the proton magnetic resonance spectroscopy (^1^H MRS). Subsequently, all spectroscopic and shimming volumes were manually placed in the selected brain regions (outside of B_0_ distortions visible on the B_0_ map). The spectroscopic data from the hippocampus [gyrus dentatus (GD), CA_1–3_] were obtained in one acquisition using the chemical shift imaging (CSI) method (Figure 1). The 2D CSI measurement was performed within 16 min with PRESS (point resolved spectroscopy) pulse sequence; 8x8 voxel matrix and 8 × 10 × 2 mm^3^ nominal voxel size (2.75 × 2.75 × 2 mm^3^ real voxel size); 22 × 22 mm^2^ FOV; TR/TE = 1500/20 ms; 36 averages and 6 kHz acquisition bandwidth. Eddy currents and B_0_ drift compensations, as well as OVS (outer volume suppression) and VAPOR (variable pulse power and optimized relaxation delays) suppression, were initiated. Linear and second order shims were automatically adjusted with the cuboid shim volume. The average linewidth of water peak was 14.8 ± 1.9 Hz.

The MRI volumetric analysis of the hippocampus was performed in ITK-SNAP (Version 3.4.0, US National Institutes of Health, Bethesda, MD, USA) software on 12 consecutive coronal T_2_-weighted MRI slices (Figure 2). The selected brain tissue volume was calculated automatically after the manual target region overlaying. Moreover, we calculated the hippocampal tissue normal volume threshold as the volumetric mean value calculated in the control animal group minus its standard deviation. Therefore, it was possible to define the percentage change in volume of the hippocampal tissue in Met-C, Met-IR-3d and Met-IR-7d animal groups with respect to the normal volume threshold.

During the spectroscopic data assessment, one representative voxel from hippocampus (Figure 1) was selected. All ^1^H MRS data curve fittings were performed by LCModel software (version 6.3-1K; S. Provencher, Oakville, ON, Canada). Subsequently, the metabolite levels of total N-acetyl aspartate (tNAA; N-acetyl-aspartate, N-acetyl-aspartyl-glutamate), myo-Inositol (mIns), choline (tCho; phosphatidylcholine, glycerophosphatidylcholine, acetylcholine, choline) and creatine (tCr; creatine, phosphocreatine) containing compounds were expressed as following ratios: tNAA/tCr, tCho/tCr, mIns/tCr and mIns/tNAA.

### 2.7. Image Analysis

Images of the coronal brain sections from the first experimental group were exported in tiff format. Images were evaluated using ImageJ software (NIH, Bethesda, MD, USA). Firstly, red, green and blue (RGB) channels were converted to 8 bit grey-scale images. Threshold levels were set from 13 pixels (min) to 255 pixels (max). Particle analysis was finished based upon size restrictions of 0 mm^2^-infinity leaving morphology unspecified. A total of 534 fields of view were analysed. The total number of Fluoro-Jade C+ (green fluorescent cytoplasm), NeuN (red fluorescent cytoplasm and processes), GFAP (green fluorescent cytoplasm and processes) was directly counted in the sections of the hippocampal CA1 area of each staining, respectively (2–3 sections per animal). The sampling grid size was set up to: 6 × 6 mm for each area. All calculations are expressed as the total number of labelled cells per mm^2^.

### 2.8. Statistical Analysis

Data acquired from image analysis of brain sections were analysed using GraphPad Prism software, version 6.01 for Windows (La Jolla, San Diego, CA, USA). The data were evaluated using one-way and two-way analysis of variance (ANOVA). Statistical analysis for spectroscopic and MRI volumetric data was accomplished using the SPSS software package (Version 15.0; Chicago, IL, USA). The differences in hippocampal volumes and metabolite ratios between animal groups (naïve control, Met-C, Met-IR-3d and Met-IR-7d) were analysed by independent-samples 2-tailed *t*-test.

## 3. Results

### 3.1. Determination of Plasma Homocysteine Level

Analysis of plasma level of Hcy in animals has shown that total plasma Hcy level in animals with 28 days of Met diet (Met-C) was significantly higher when compared to the naïve male control Wistar rats (7.08 ± 0.33 μmol/l, *n* = 5), and reached 11.22 ± 3.86 μmol/l (*n* = 5). 

### 3.2. Histo-Morphological Changes in the Rat CA1 of Hippocampus 

#### 3.2.1. FluoroJade-C (FJC) Staining

To show the extent of neurodegeneration, FJC was used for detection of disintegrated neurons in the tissue slices of the CA1 area of the rat hippocampus. As expected in both control groups, we could not detect any FJC positive neurons (Figure 3). Unsurprisingly, we found a 47-fold increase in the neuron positivity in the ischemic group followed by 3 days of reperfusion (IR-3d; *p* < 0.001), in comparison to both controls (C and Met-C). Interestingly, in the same group with the Met diet (Met-IR-3d), the differences reached 58.33-fold increase (*p* < 0.001; Figure 3), compared to both controls and rose up to 24% (*p* < 0.01), when compared to the naïve ischemized group. 

In the prolonged reperfusion period (7 days) in both groups of animals (naïve and Met diet treated), we documented persisted increase of FJC+ neurons (32.21-fold (*p* < 0.001), resp. 21-fold (*p* < 0.001) compared to both control groups (naïve and Met treated) and differences between naïve ischemized and Met ischemized groups represented 36.4% (*p* < 0.01). Remarkably, longer reperfusion time led to an apparent reduction of the cellular FJC positivity representing 32.3% (*p* < 0.001), resp. 44.6% (*p* < 0.001), when compared to the naïve IR-7d and IR-3d groups and Met-IR-3d group, probably as a result of an extensive tissue breakdown. This can be seen by the morphologically changed neurons (arrowhead), as well as scarred, unstained neuronal tissue which substitutes neuronal loss (asterisk), likely as a result of prolonged ischemia and neuronal tissue remodelling (Figure 3). 

#### 3.2.2. NeuN and Glial Fibrillary Acidic Protein (GFAP) Measurements

NeuN and GFAP measurements were used for the detection of the histo-morphological changes in neuronal cells and astrocytes as well as for the quantification of the cell numbers. 

In the naïve control group (C), NeuN antibody labelled nuclei as well as the cytoplasm of the majority of neuronal cell types of all regions in the adult brain (cerebral cortex, hippocampus and cerebellum). No immunoreactivity was observed in the astrocytes, neither in the nuclei nor the cytoplasm. The preponderance of the cytoplasmic immunopositivity was concentrated in the soma, extending to a short distance into the processes (mainly axon hillock; Figure 4—arrow). 

In the control Met group (Met-C, Figure 4), we detected morphological changes in neurons of CA1 (swelling of the body as well as nuclei—arrowhead), with no statistically significant differences in numbers, when compared to the naïve controls. 

In the naïve ischemized group with 3 reperfusion days (IR-3d), we detected 36.7% (*p* < 0.01) decrease in the number of NeuN+ neurons in comparison to the naïve control. In the same group with the Met diet, we found a statistically significant decrease of NeuN+ neurons in comparison to the naive controls to 44.9% (*p* < 0.001) as well as to 51.4% (*p* < 0.001) compared to the Met controls (Figure 5). Additionally, we detected morphological changes, such as a loss of staining from processes, reduction of positivity in soma, nuclei and a decrease in intensity of labelling, however, the changes were not statistically significant neither in the fluorescence intensity nor in the number of NeuN+ cells. 

The tendency of diminution of the NeuN cell positivity seems similar also in the prolonged ischemic period at 7 reperfusion days. We observed the highest decrease in cell positivity, up to 39.2% (*p* < 0.001) in the naïve ischemized group, as well as in the ischemized Met group up to 49.2% (*p* < 0.001), when compared to the naïve controls and 43.1% versus Met controls (*p* < 0.001). In addition, Met diet induced diminishing in the number of maturate neurons between naïve and Met group up to 31.7% (*p* < 0.05; Figure 5) with successive, deeper morphological changes and focal areas with neuronal loss as well as lack of fluorescent intensity (Figure 4—asterisk).

Immunolabelling of GFAP in the naïve controls (C) presented astrocytes with radially arranged thin processes. Astrocytes were dispersed in all layers of the hippocampus (Figure 4—arrow). In the Met controls (Met-C), we ascertained a remarkable increase of astrocytic GFAP expression predominantly in the stratum oriens, stratum radiatum, and stratum lacunosum. We also found a different morphology of astrocytes in this group, probably as a result of hHcy impact on astrocytic activation (Figure 4). The number of astrocytes were elevated by 24.8% (*p* < 0.01), in comparison to the naïve control (Figure 4). 

Similar to our previous experiments, we detected an increase of astrocytes in the naïve ischemized IR-3d group 24% (*p* < 0.01) when compared to the naïve controls. However, in the same Met ischemized group, surprisingly, we found a diminished number of astrocytes (Figure 4) up to 18.8% (*p* < 0.01) versus naïve control, and 34.4% (*p* < 0.001) versus Met-C (Figure 5). Similarly, we saw only moderate changes in the astrocytes’ morphology (prolonged processes, wider soma), and the reduction in the intensity of the fluorescent signal was distinct.

Remarkable patterns of astroglial activation were also seen in naïve ischemized group after 7 days of reperfusion, even though we did not find any statistical differences in the number of GFAP positive astrocytes when compared to the all of the other experimental groups. Moreover, hypertrophic somas with thicker, flagging and branched processes of astrocytes were visible, which indicate the activation of astrocytes (IR-7d; Figure 4). Striking histo-morphological alternations were also displayed in the Met ischemized group at 7 reperfusion days, even when compared to its naïve counterpart with the characteristic flagging, shorter and thinner astrocytic processes. We found a diminished number of GFAP positive cells of this group in comparison to the Met controls in 15.4% (*p* < 0.001). In addition, an extensive elevation was detected when compared to the 3 days reperfusion group, up to 28.9% (*p* < 0.01; Figure 5), even though there were no statistical differences between Met pre-treated and naïve animals after 7 reperfusion days.

### 3.3. ^1^H Magnetic Resonance Spectroscopy (MRS) Analysis and Volumetry

In this part, we describe results of ^1^H MRS analysis and volumetry on the ischemized naïve as well as the Met pretreated groups. 

Quantification of ^1^H MRS in the hippocampus of ischemized animals (Met-IR-3d and Met-IR-7d) showed a significant decrease in tNAA/tCr ratio, and a significant increase in mIns/Cr when compared to the non-ischemized (C and Met-C) groups (Table 1). However, there were no significant differences between naïve controls and Met-C as well as Met-IR-3d and Met-IR-7d groups. Furthermore, we detected an increase in mIns/Cr in animals 7 days post-ischemia in comparison to the controls, Met-C, as well as to 3 days post-ischemia. Analysis of tCho/tCr showed no statistical significance between all analysed stages (Table 1).

MRI volumetric analysis found significant changes in the hippocampal volumes between naïve controls and Met-C groups (*p* = 0.031). The hippocampal volume was significantly increased in Met-IR-3d and significantly decreased in Met-IR-7d group, compared to the naïve control (C) as well as to the Met-C groups (Table 2). On the 7th day after ischemia in the Met group, we also detected a significant decrement in the hippocampal volume (*p* < 0.001) when compared to the 3rd post-ischemic day. With respect to the naïve control volume threshold level, we found an increased level in Met-C (10 ± 2%) and in the Met-IR-3d (23 ± 6%) animal groups, however, in the Met-IR-7d group (−6 ± 2%) was confirmed hippocampal volume decrement (Table 2).

## 4. Discussion

### 4.1. Histo-Morphological Changes in the Hippocampus after Met Diet Induced Hyperhomocysteinemia (hHcy) and Effect of IRI 

High Met protein intake or dysregulation of the Met metabolism can potentially lead to hHcy, as a known risk factor for cardio- and cerebrovascular disorders [14,15], associated with the propensity to the development of neurodegeneration linked with dementia, Parkinson or AD [1,3,5], but the exact mechanism of Hcy involvement is not fully clear. As shown earlier by several studies, cerebral tissue undergoes massive time-dependent heterogeneous histopathological changes after the ischemic stroke [20,21,22,23,24,27,28]. The hippocampus is especially vulnerable to the ischemic damage at the early stages of neurodegenerative processes [28,29]. In this paper, we applied a novel methodological approach, which uses 1H MRS and MRI volumetry, and an immunohistological analysis, to bring more light into the pathomechanisms associated with the global cerebral ischemia and reperfusion in combination with the model of Met diet-induced hHcy.

Despite the high clinical significance, only a limited number of experimental approaches can be found in the literature to describe the mutual influence of co-morbid hHcy to ischemic damage in animal models which mimic a human ischemic stroke [30]. Based on our previous studies and works from other laboratories, the presumptive mechanism of neurodegeneration progression in the combined hHcy and IRI conditions are shown in the Scheme 1 and in detail described in Lehotsky et al. [5,31].

If the plasma levels of Hcy exceed 10–15 μmol/l, there is a condition known as hHcy, which occurs as a result of an inborn error of methionine metabolism or by non-genetic causes [35]. In our experiment, in the Met-C group was the level of Hcy increased 1.5-fold in comparison to the naïve control animals that matches the criteria for mild hHcy [35,36]. Dos Santos et al. [37] demonstrated in their hHcy model (subcutaneous injection of DL-Hcy in the lower dose of 0.03 μmol/g of the body weight of male Wistar rats twice a day for 30 days that this dose increases rats Hcy plasma concentration, similarly described in the moderate hHcy patients. More importantly, Nuru et al. [38] demonstrated that high Met diet (1.2%), low folate (0.08 mg/Kg) and low vitamin B6/B12 (0.01 mg/Kg; 10.4 μg/Kg) lead to significant neuronal damage which, interestingly precedes vascular dysfunction. 

Met diet-enriched hHcy conditions induce neuronal damage in the hippocampal region, as we determined earlier [20]. Our study here proved that the extent of the degeneration process clearly aggravates, if hHcy rats were subjected to the subsequent IRI. The hippocampus manifested an increased number of disintegrated neurons, concomitant with the extensive changes in the glial cellularity. Remarkably, the neurodegenerative patterns worsen with the prolonged reperfusion times and the apparent diminishing in FJC+ cell at the 7th post-ischemic day is probably the result of the comprehensive neuronal disintegration, as we observed in our previous studies [20,21,22,23,24]. Our results are also consonant with the results of other laboratories [6,27,39], as well as clinical observations [6,40,41] and supports the idea that Met-enriched diet initiates hippocampal neurodegeneration which is aggravated after the global IRI (Figure 4).

Astrocytes are highly plastic cells and their dynamic morphological changes could affect the intercellular communication with surrounding synapses that are important in the development of brain lesions [42]. The increasing data suggests that astrocyte death can precede neuronal loss and aggravate ischemia-induced brain injury [43]. Lukaszevicz et al. [43] referred to that in contrast with the commonly held assumption that astrocytes are the most resistant cells to ischemic conditions. Protoplasmic astrocytes (grey matter) may indeed suffer altered cerebral blood flow rapidly and irreversibly. In our previous studies [21,34] we used the TUNEL method for the detection of cell death in the rat brains. In the naive animals 3 days post-ischemia, we did not find any marks of astrocytes positivity. In the work of Kovalska et al. [21], we found TUNEL positivity in neurons as well as in astrocytes in hHcy animals with/without ischemia mainly in the brain cortex. Furthermore, in our previous paper [20], we demonstrated that Met diet in rats induces an increased number of astrocytes in the hippocampal area. In this study, Met rats subjected to the IRI manifested paradoxically a diminution of astrocytes level after 3 days of reperfusion. However, we found in this group a hypertrophic soma with flagging, short, thicker and branched processes of astrocytes that indicate the hypertrophy of astrocytes [44,45]. Activation of astrocytes was referred to as a possible mal-adaptive alteration in neuronal functions [45] since astrocytes proliferation reflects the early stages of the most neurodegenerative processes. Similarly, in AD linked experiments, moderate to severe diffuse astrogliosis is considered a well-known hallmark of diseases, such as AD [44,45]. Weekman et al. [46] showed in their in vitro study that astrocytes after treatment with a mild level of Hcy, manifested decreased levels of several astrocytic end feet genes at 72 h post-ischemia. Maler et al. [47] reported that Hcy level of 2 µmol/l and above, induced a dose-dependent cytotoxic effect on protoplasmic astrocytes. Astrocytes regulate expression of the neuronal N-methyl-D-aspartate (NMDA) receptor subtypes, which increase neuronal sensitivity to glutamate toxicity and thus accelerate the initial step in the program of reactive astrogliosis and dynamics of astrocyte response to the damage [5]. Cervetto et al. [48] demonstrated that increased production of Hcy at the astrocyte level may lead not only to the activation of the NMDA receptors, but might also act on the glutamatergic transmission by reducing dopamine D2 receptor signalling, therefore facilitating the release of glutamate from astrocytes. Stimulation of NMDA receptors by Hcy increases Ca2+ influx that exerts neurotoxic effects [49]. 

However, 7 days of post-ischemia, the number of astrocytes elevated but manifested significant signs of impairment such as less, flaccid processes in comparison to the naïve IR-7d group as well as Met-IR 3d animals. We presume that the changes at 3 days presented a subacute, partially reversible stage of glia disintegration (linger oedema, reduction of grey matter) or dysregulation in neurogenesis, as a reaction to the mild hHcy conditions [50]. Based on our previous results and outcomes from this experiment (decreased number of GFAP+ astrocytes as well as a considerable decrease in the metabolite ratio of mIns/tCr), and according the aforementioned pieces of knowledge (activation of NMDA receptors, increase of Ca2+ influx, cytoskeletal remodelling and the function of astrocytes as a glutamate “trapping” cells), we assume that hHcy enriched by Met diet in combination with IRI 3 days post-ischemia lead to a subacute, partially reversible changes in the number of astrocytes. Additionally, this study also indicates that different types of hHcy induction (direct Hcy subcutaneous administration or by Met enriched diet), could lead to a different time-dependent level of histological damage. Apparently, the level of Hcy in plasma, or its metabolic conversion to the toxic metabolites in the tissues might play a significant role in the triggering noxious conditions. We suggest that the Met diet in combination with ischemic insult in in vivo conditions initiates parallelly neuronal disintegration as well as the astrocytic impairment.

### 4.2. In Vivo Metabolic Changes in the Brain after Met Diet Induced hHcy and the Effect of Ischemia-Reperfusion Insult (IRI)

In vivo ^1^H MRS and MRI-volumetry analysis were applied as a non-invasive approach to determine the metabolic ratio and alternations in the hippocampal volume in ischemized animals with the Met diet. 

As generally accepted, tCr emerges as a constituent of all brain cells. tNAA is considered to be the characteristic neuroaxonal marker, tCho and mIns are regarded as the glial components, e.i. tCho (mainly for oligodendrocytes) and mIns for astrocytes [51,52]. 

The results of ^1^H MRS revealed changes in all selected metabolites in the hippocampal area (Table 1). In our previous paper [20], we showed that enriched Met diet (hHcy) alone is able to initiate changes in the key metabolites, which are related to the incipient hippocampal neurodegeneration. In this study, we demonstrate that IRI, when combined with the Met diet, induces secondarily more severe pathobiochemical and metabolic impairments. Those are morphologically manifested by oedema, neuronal tissue breakdown and glial impairment. Notably, extend of the insult could vary among the animals, which might lead to higher volatility in the determined levels of metabolite in the animal groups.

Reduced tNAA level is generally interpreted as a parameter of neuronal/axonal/dendritical density, dysfunction and/or loss, which is a characteristic feature for ^1^H MRS [41,51,52]. Preclinical investigations have shown that tNAA decreases in the ischemic brain parenchyma in a linear fashion for the first 6 h, followed by a slower decrease for the subsequent 24 h [53]. On the other hand, there is spare information on how in vivo neuronal tissue can metabolically cope with the hHcy conditions. In this study, we found a significant time-dependent decrease of tNAA/tCr ratio, with the most severe decrement at the 7th reperfusion day. This is a consonant to the histological analysis, which shows an inhibition and the loss of post-mitotic neurons.

A precursor of membrane metabolism, Cho, is considered to be a marker of membrane density, i.e., phospholipids synthesis and degradation [54]. Increased tCho relative to tCr has been attributed to the cerebral infarctions, inflammation and multiple sclerosis [51,55]. It was recently shown that tCho was associated with the membrane turnover, which was directly related to Hcy removal [54], and a link between hHcy and ceramide metabolism was revealed in the AD-type neurodegeneration [52,56]. In our experiments, we did not find significant changes, neither in the levels of tCho nor the tCr, at any reperfusion time. However, the rise in its ratio suggests the proposed process of hippocampal re/de-remyelination, alter neuroglial homeostasis, and cell membrane turnover in the different reperfusion times. 

Decreased mIns was observed early after trauma BI [57,58], which may reflect mIns efflux from astrocytes, as a volume-regulatory factor under conditions of oedema, or alternatively, it may reflect cell lysis and death [59]. By contrast, pathologically activated astrocytes with larger cell volumes tend to have an increased mIns levels [60]. In fact, in the progression of the injury (several days to weeks), an increased mIns was suggested as a sign of neuroinflammation and reactive astrogliosis [58,61]. A link between increased mIns and astrocyte activation was also shown in animal models of status epilepticus [62], in AD, trauma brain injury [63], and also in clinical studies in patients with frontotemporal lobar degeneration [64] and aging [61]. In this work, we found an increased ratio of mlns/tCr in Met diet-treated animals 7 days post-ischemia, suggesting changes in the hippocampal astrocytes. Regardless, lmIns is a precursor of phospholipid membrane constituents and its concentration is affected by the formation and breakdown of myelin. Therefore, the higher level of mlns could also reflect the process of de-myelination in our experiment [51,52,61]. 

Collectively, results of metabolic MRS analysis indicate that IRI in animals with Met-enriched diet initiates progressive metabolic disturbances with the dysregulation of myelinated tract in the hippocampus. This information might have clinical relevance in terms of human hHcy conditions linked with ischemic stroke.

### 4.3. Volumetric Changes in the Brain after Met Diet-Induced hHcy

It has been confirmed that hHcy promotes conditions, which might significantly affect tissue metabolite ratio [4,5]. In addition, the Met-Hcy cycle is connected to the one-carbon unit metabolism and editing processes of proteosynthesis. Consequently, Hcy and its metabolites might interfere with the epigenetic control of gene expression, as one of the underlying pathological factors [65]. Epigenetic dysregulation mediated by hypo/hypermethylation of DNA or changes as histone N-homocysteinylation might have a detrimental effect on the metabolic disparities in the hHcy conditions. 

Our previous work, Kovalska et al. [20] demonstrated that hHcy alone, induced by Met diet, leads to the characteristic increase in the hippocampal volume. IRI in these conditions is manifested by the preservation of oedema, and an additional increase at the 3rd day and subsequent decrease at the 7th day of reperfusion. This is a novel pattern, which has not yet been characterized in naïve (not hHcy) ischemic animals [27,66,67]. Sustained oedema might be the result of the blood–brain barrier (BBB) disruption caused by the elevated Hcy or its metabolites, as well as a consequence of the direct excitotoxic effect of Hcy on astrocytes [16]. Most of the BBB damage usually occurs prior to 48 h post-stroke, as shown in the embolic ischemic model in rats [27,66]. It is known that hHcy can impair the endothelium’s capacity to regulate vascular tone by reduced bioavailability of NO, which leads to the endothelial dysfunction [6,68]. This could be one of the most important mechanisms by which hHcy exerts endothelial damage including also a greater sensitivity in cerebral microvessels [35].

Early experimental data have shown that a moderate hHcy induces cardiac remodelling and reduces cardiac performance. Studies also suggest that even a mild hHcy (10–15 μmol/l) can impair cardiac contractility, alter substrate metabolism, and cause cardiac remodelling and dysfunction [69]. Another paper from this laboratory [70] showed that a chronic mild hHcy (13.2 ± 1.2 μmol/l) resulted also in the significant deterioration of cardiac function, characterized by systolic and diastolic dysfunction linked with an increase of mitochondrial lipoperoxidation, activation of endoplasmic reticulum stress and releasing of reactive oxygen species (ROS) into the extramitochondrial sites [70]. The ROS caused by hHcy can react with biological molecules that can lead to the irreversible alternations of their ultimate functions (sometimes even rendering them completely inactive). HHcy state abnormally activated MMP-9, degraded the components of the extracellular matrix as well as the gap junction protein connexin-43, thereby causing fibrosis and vascular dysfunction [71]. The other potential mechanism is a structural protein modification due to the N-homocysteinylation which presents a covalent modification of Hcy thiolactone and results in the protein denaturation, enzymatic inactivation and even amyloid formation that inhibits endothelial function, thus establishing a vicious cycle for impairing brain circulation [33] that is associated with neurodegenerative diseases [35]. The aforementioned changes could lead to the deregulation of adjacent astrocytes. Astrocytes have been shown to modulate extracellular space volume as well as the neurotransmitter homeostasis [72]. The observed decrease of the hippocampal volume at the 7th reperfusion day could be a sign of progressive neurodegeneration, manifested by the decrement of the cell population in the hippocampus. These findings are in accordance with previous studies which describe the progression of neurodegenerative processes [61,64,66,67].

Based on these results, we suggest that Met-enriched diet inducing a mild form of hHcy in combination with IRI in rats represents an experimental model, which can evolve a toxic body environment with potentially detrimental impact to the neuronal tissue, seen on the morphological and metabolic levels. This is presented by the changed metabolic ratio, volume disturbances, attenuated neurites and activation of astrocytes in the rat hippocampus. 

## 5. Conclusions

In conclusion, our study utilizes a novel approach which combines histological analyses with the non-invasive in vivo 1H MRS, MRI methodology, with the aim to reveal the potential histo-morphological as well as the metabolic hippocampal dysregulation in rats with Met diet subjected to IRI. Both methods showed that hippocampi of ischemized animals with Met diet manifested exaggerated neurodegeneration, mainly in the selectively vulnerable CA 1 hippocampal region. Evidence of a possible causal link between the ischemic insult with hHcy induced by the Met diet and incipient neurodegenerative processes would contribute to better understanding of the potentially deleterious impact of both modifiable risk factors (hHcy and IRI) in clinically relevant neurological disorders. 

## 6. Limitations of Study

The limitation of this study was in the measurement of metabolites because their level could fluctuate in chronic conditions due to the occurrence of cytotoxic oedema (tissue water), which could result in a lack of correlation between ^1^H MRI and histology Furthermore, there are different areas in the hippocampus with different vulnerability to acute or chronic insults that could influence the overall metabolic ratio. Our results support these findings and the need for a multiparametric MRI approach, in combination with histologic analysis to characterize the complex evolution of neurodegenerative diseases [73].

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
