# Peer review of "Effect of Methionine Diet on Time-Related Metabolic and Histopathological Changes of Rat Hippocampus in the Model of Global Brain Ischemia"

_biomolecules, 2020, doi:10.3390/biom10081128_

Round 1

Reviewer 1 Report

COMMENTS 1:

Serum Hcy below 15μmol/l, strictly speaking, does not meet the criteria of hHcy. In the present study, the authors showed moderately elevated Hcy alone is able to initiate hippocampal neurodegeneration. This result suggests Hcy may cause adverse effect on neurons at lower levels than in cardiovascular system. Should the authors give more discussion on this result?

COMMENTS 2:

There were no further results suggesting the possible mechanisms of Hcy aggravating IRI process. Should the authors add more supplementary materials explaining the mechanisms?

Author Response

See enclosed file

Reviewer 2 Report

The authors investigate the influence of hyperhomocysteinemia on ischemia-reperfusion insult (IRI) induced brain injury in rat Hippocampus. They find that methionine diet induced hyperhomocysteinemia aggravates IRI induced degeneration of the hippocampal neurons.

Their findings are interesting. However, there are some questions need to be addressed by the authors.

1, the fluorojade C stains can visualize disintegrated neurons. However, IRI can induce neuron death and apoptosis in brain. Therefore TUNEL staining should be used to visualize apoptotic neurons induced by IRI. The authors find the decreased astrocyte in met groups on IR-3d. They assume that IRI plus Met may induce astrocyte degeneration as well. However, no direct evidence can support the hypothesis. Therefore if TUNEL staining can be applied to their study, astrocyte under apoptosis can be detected in their brain slices.

2, the authors find decreased astrocyte number under Met plus IRI. They suspect that IRI plus met aggravate astrocyte degeneration in brain. However, the supporting evidence is lacking. The authors should provide more evidence.

Author Response

See enclosed file

Round 2

Reviewer 2 Report

The manuscript has been significantly improved after revision.

Author Response

Thank you very much for your response and comments